# TCA cycle tailoring facilitates optimal growth of proton-pumping NADH dehydrogenase-dependent *Escherichia coli*

Nikita Goel,[1] Stuti Srivastav,[1] Arjun Patel,[2] Akshay Shirsath,[1] Tushar Ranjan Panda,[3] Malay Patra,[3] Adam M. Feist,[2,4] Amitesh Anand[1]

**ABSTRACT** The bacterial lifestyle is plastic, requiring transcriptional, translational, and metabolic tailoring for survival. These dynamic cellular processes are energy intensive; therefore, flexible energetics is requisite for adaptive plasticity. An intricate network of complementary and supplementary pathways exists in bacterial energy metabolism. There are two main entry points for electrons in the aerobic electron transport system, NADH dehydrogenase (NDH) and succinate dehydrogenase (SDH), receiving electrons from NADH and succinate, respectively. Aerobic bacterial phyla have a non-proton-pumping NADH dehydrogenase, which is often the primary dehydrogenase under aerobiosis. Here, we report adaptive changes supporting growth restoration in an *Escherichia coli* strain lacking the primary dehydrogenase. Growth optimization is achieved by reducing the activity of succinate dehydrogenase, and thus we demonstrate a physiological discord between proton-pumping NADH dehydrogenase and succinate dehydrogenase in supporting growth. Beyond the fundamental understanding of the bioenergetic network, identifying this compensatory feature provides impetus to rational antimicrobial combinations for targeting the non-proton-pumping dehydrogenase.

**IMPORTANCE** Energy generation pathways are a potential avenue for the development of novel antibiotics. However, bacteria possess remarkable resilience due to the compensatory pathways, which presents a challenge in this direction. NADH, the primary reducing equivalent, can transfer electrons to two distinct types of NADH dehydrogenases. Type I NADH dehydrogenase is an enzyme complex comprising multiple subunits and can generate proton motive force (PMF). Type II NADH dehydrogenase does not pump protons but plays a crucial role in maintaining the turnover of NAD+. To study the adaptive rewiring of energy metabolism, we evolved an *Escherichia coli* mutant lacking type II NADH dehydrogenase. We discovered that by modifying the flux through the tricarboxylic acid (TCA) cycle, *E. coli* could mitigate the growth impairment observed in the absence of type II NADH dehydrogenase. This research provides valuable insights into the intricate mechanisms employed by bacteria to compensate for disruptions in energy metabolism.

**KEYWORDS** bioenergetics, metabolism, adaptive laboratory evolution, electron transport

The evolutionary repurposing of biochemical pathways and enzyme complexes has resulted in the emergence of a highly efficient energy generation system in bacteria. The origin of membrane-resident oxidative phosphorylation (OXPHOS) can be traced back to primitive fermentative metabolism (1). Later, the oxygenation of the earth's environment is believed to ascribe additional responsibility to the tricarboxylic acid (TCA) cycle to facilitate the operation of the most efficient electron transport system (ETS) for driving OXPHOS, wherein NADH is the electron donor and oxygen is the terminal

Address correspondence to Amitesh Anand, amitesh.anand@tifr.res.in.

Nikita Goel and Stuti Srivastav contributed equally to this article. Their authorship order was determined alphabetically.

The authors declare no conflict of interest.

See the funding table on p. 5.

electron acceptor (2, 3). This TCA-OXPHOS network has become the hallmark of modern cellular energetics.

The reducing equivalents generated by the central carbon metabolism transfer electrons to the ETS primarily through NADH dehydrogenase (NDH-1) and succinate dehydrogenase (SDH). The evolutionary rise in the metabolic complexity, and consequent demand for faster regeneration of pyridine nucleotides, potentially led to the emergence of high turnover NADH dehydrogenase (NDH-2), which is preferred over NDH-1 by many aerobically growing bacteria (4, 5). The oxidation of NADH by NDH-2 is not coupled to periplasmic proton translocation and hence does not directly participate in proton motive force (PMF) generation. However, NDH-2 can support a higher metabolic flux and occupies a lesser membrane volume (5).

Here, we examine the proximal and distal consequences of the lack of NDH-2 in *Escherichia coli* by performing adaptive laboratory evolution (ALE) of the Δ*ndh* strain, followed by multi-omics and biochemical analyses. We observe (i) adaptive restoration of the growth defect in the Δ*ndh* strain, (ii) loss of function of SDH facilitating aerobic growth improvement, and (iii) a potential antagonism between NDH-1 and SDH in *E. coli* Δ*ndh* strain. Notably, NDH-2 is vital for several pathogens while being absent in mammalian mitochondria (6, 7). Therefore, this enzyme holds clinical significance. Besides delineating the fundamental design principle of bioenergetics, the identification of this adaptive axis will facilitate the efficient targeting of this enzyme for therapeutic purposes.

*E. coli* ETS consists of both types of NADH dehydrogenases allowing conditional coupling of the electron flux to PMF (Fig. 1A) (8). NDH-1 consists of 13 subunits, whereas NDH-2 is a monomeric flavoprotein. The $K_m$ of NDH-2 is greater than NDH-1, so a higher NADH oxidation rate can be achieved (9).

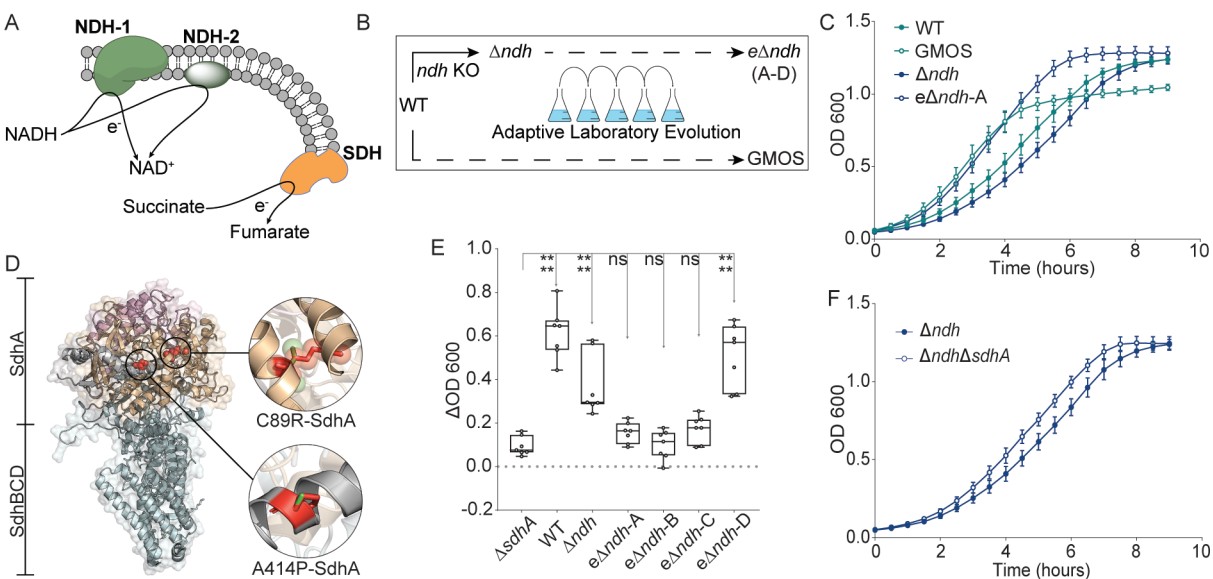

**FIG 1** Adaptive laboratory evolution (ALE) of *E. coli* Δ*ndh*. (A) NDH-1 and NDH-2 accept electrons from NADH, and succinate dehydrogenase (SDH) accepts electrons from succinate. The electrons then enter into the ETS. (B) A scheme for strain generation and ALE design for this study. (C) The growth profile of the unevolved and evolved strains. The wild-type (WT) strain evolved on M9 minimal medium with glucose as the carbon source (GMOS) is included as a reference for the expected maximum growth profile of *E. coli*. (D) The succinate dehydrogenase mutations acquired during adaptive laboratory evolution are mapped on the SDH structure from *E. coli* (Protein Data Bank (PDB) entry 1NEK). The truncated segment is shown in pink; the FAD-binding domain of the SdhA is shown in golden brown; the remaining region of SdhA is cyan. Cofactors and ligands are in dark gray; SdhBCD subunits are in gray. WT and mutated residues are shown in green and red sticks, respectively. (E) *In vitro* SDH activity of Δ*sdhA*, WT, and Δ*ndh* (unevolved and evolved) strains. All seven replicate values are displayed, and whiskers represent the maximum and minimum data points while the solid line inside the box plots represents the median. A significance test using one-way analysis of variance (ANOVA) was performed compared to Δ*sdhA*. (F) Effect of *sdhA* gene deletion on the growth of Δ*ndh* strain. The growth curves in panels (C) and (F) represent a mean of five biological replicates (with three technical replicates each), and the error bars show the standard error of the mean. The carbon source for the growth was glucose.

There are conflicting reports on the relative contributions of the two NADH dehydrogenases during aerobic growth (10, 11). We genetically engineered an *E. coli* strain lacking NDH-2 (Δ*ndh*), wherein we observed a growth defect as compared to the wild-type strain (Fig. 1B and C). However, the bacterial ETS is highly flexible, and therefore we performed adaptive laboratory evolution to examine the possibility of growth optimization in this strain—four independent lineages were evolved (Fig. 1B; Table S1). In about 300 generations, the Δ*ndh* strain achieved a growth rate similar to the wild-type strain that evolved on M9 minimal medium containing glucose as the carbon source (Fig. 1C; Fig. S1A) (12, 13).

Distal adaptive responses are often facilitated by genetic changes; we, therefore, examined the genome sequences of the evolved strains and compared them with unevolved strains. Interestingly, all four independently evolved lineages of Δ*ndh* acquired a mutation in TCA cycle enzymes (Table S2). Three lineages mutated subunit A of the SDH (*sdhA*), which provides a direct redox link between cytoplasm and membrane (14). One of the mutations in *sdhA* generated a premature termination codon, indicating the loss of function of this enzyme. Other point mutations resulted in two SdhA protein variants, C89R-SdhA and A414P-SdhA (Fig. 1D). The C89R variant is located in the FAD-binding domain, and the A414P lies between the FAD-binding domain and the fumarate reductase/succinate dehydrogenase flavoprotein-like, C-terminal domain of the SdhA subunit. The C89R variant introduces a charged residue into a buried region of the protein, which can negatively impact protein folding, and the guanidinium side chain can lead to steric clashes with residues of the neighboring helix. The other variant, A414P, leads to the introduction of proline in the middle of the 25-residue helix region. The lack of amide proton in proline precludes it from participating in hydrogen bond formation and thereby disrupts the α-helix when not located at one of the first three positions of that helix (15–17).

We experimentally verified the effect of the mutations by performing an *in vitro* SDH activity assay. We observed a drop in the SDH activity in all evolved Δ*ndh* (eΔ*ndh*) lineages with *sdhA* mutation (Fig. 1E). SDH activity is critical for *E. coli*'s growth on succinate; therefore, we performed growth characterization of the strains on M9 minimal medium containing succinate as the sole carbon source (Fig. S1B). As expected, the Δ*sdhA* strain failed to grow on succinate. The growth retardation of Δ*ndh* was more pronounced on succinate, which could be due to the non-fermentable nature of this carbon source. eΔ*ndh* lineages with *sdhA* mutation showed drastic growth retardation. eΔ*ndh*-A showed a complete growth failure on succinate. Notably, this lineage has the truncated SdhA protein.

To probe the causal nature of *sdhA* mutations, we knocked out *sdhA* in the Δ*ndh* strain. While *sdhA* deletion did not affect the growth profile of *E. coli* on glucose (Fig. S1C), Δ*ndh*Δ*sdhA* showed improved growth as compared to Δ*ndh* (Fig. 1E). The eΔ*ndh* strains have higher growth than Δ*ndh*Δ*sdhA*, potentially due to the other mutations acquired during ALE. eΔ*ndh*-B and eΔ*ndh*-D acquired additional mutations in a well-characterized genetic region *rph-pyrE*, potentially relieving a defect in pyrimidine biosynthesis caused by a 1-base deletion in the *rph-pyrE* operon (13, 18, 19). The other lineages, eΔ*ndh*-A and eΔ*ndh*-C, deleted 103 nucleotides from within a putative RNase adaptor protein, *yicC*. We verified the impact of this mutation by deleting *yicC* in wild type, Δ*ndh* and Δ*ndh*Δ*sdhA*. We observed that *yicC* deletion improves the growth of Δ*ndh*Δ*sdhA* strain (Fig. S1D and E). YicC regulates a small regulatory RNA, RhyB, that modulates the central carbon metabolism and redox homeostasis (20, 21). However, *yicC* is immediately upstream of the *rph-pyrE* region, and the deletion may be improving the growth through the optimization of *pyrE* expression.

Furthermore, we probed whether there is any subunit-specific contribution of SDH in improving the growth by knocking out *sdhC* in the Δ*ndh* strain. We observed a similar growth improvement in Δ*ndh*Δ*sdhC* as compared to Δ*ndh* (Fig. S1C and D), suggesting reduced SDH activity as the cause for the growth optimization of Δ*ndh* strain.

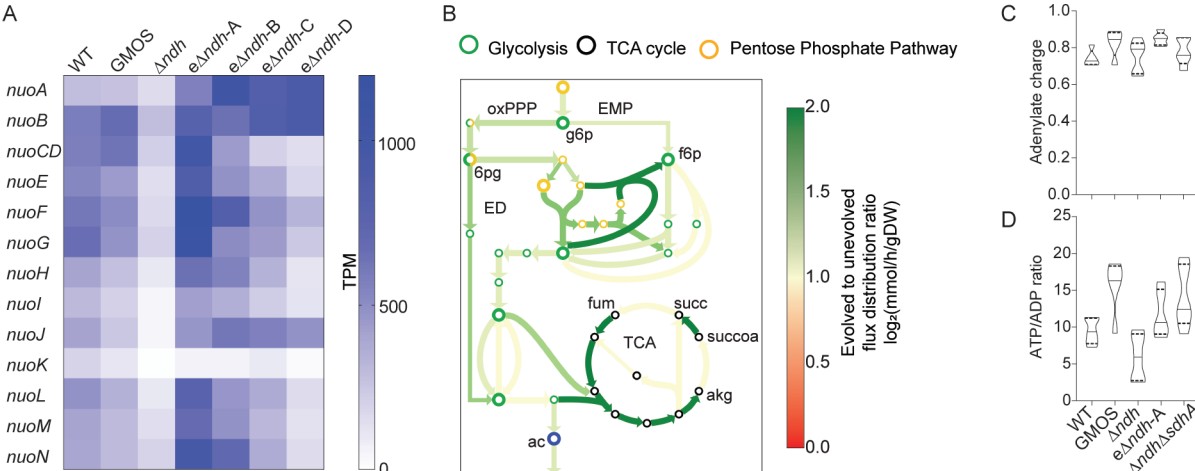

**FIG 2** Examination of adaptive rewiring during *E. coli* Δ*ndh* ALE. (A) Heatmap showing expression levels (TPM: transcripts per million) of the type I NADH dehydrogenase genes (*nuoA-N*). (B) Computed metabolic flux maps depicting the central carbon metabolism in the evolved Δ*ndh* as compared to unevolved Δ*ndh*. Metabolites are indicated in the figure as follows: g6p, D-glucose-6-phosphate; f6p, D-fructose-6-phosphate; 6pg, 6-phospho-D-gluconate; ac, acetate; succ, succinate; fum, fumarate; succoa, succinyl-Co-A; akg, alpha-ketoglutarate (oxPPP, oxidative pentose phosphate pathway; EMP, Embden-Meyerhof-Parnas pathway; ED, Entner-Doudoroff pathway; TCA, tricarboxylic acid cycle). The estimation of adenylate charge (C) and ATP to ADP ratio (D) using ultra performance liquid chromatography (UPLC). Five biological replicates were used, and the solid lines in the violin plot represent the median while the dotted lines represent the upper and lower quartiles. The strain label on the *x*-axis of panels C and D is the same.

Both NDH-1 and NDH-2 can direct electron flux from NADH to the ETS and thus can substitute for each other. We examined the transcriptional status of the genes encoding NDH-1 (*nuoA-N*) in the Δ*ndh* strain. Interestingly, the adaptive upregulation of *nuoA-N* genes was observed after adaptive evolution only (Fig. 2A), indicating the adoption of a metabolic state favoring NDH-1 operation. There appears to be a compensatory role of NDH-1 in the Δ*ndh* strain. There appeared a heterogeneity in the expression pattern of different subunits of NDH-1 in eΔ*ndh*-D, encouraging us to take a systems biology approach in rationalizing the evolutionary outcomes.

We probed the flux through the central carbon metabolism using a genome-scale model constrained with phenotypic data (Table S3) and transcriptomics data (22). The metabolic flux simulation indicated a general increase in the central carbon metabolism in the evolved strains as compared to the unevolved strain. Interestingly, while there was an overall increase in TCA flux, several reactions showed a large reduction in flux. Specifically, the flux through SDH and 2-oxoglutarate dehydrogenase diminished significantly (Fig. 2B). The cycling of NADH is critical to driving metabolic reactions and for the maintenance of normal cell redox homeostasis (23). NDH-1 has a lower turnover for NADH oxidation, and a lack of NDH-2 can perturb the NAD$^+$ to NADH ratio, resulting in a decreased oxidative metabolism (23). The selective silencing of the NADH-producing reactions could be an adaptive response to maintain a growth-conducive redox environment.

Notably, the lactate production by the unevolved Δ*ndh* went down after evolutionary optimization suggesting an improved respiratory capacity (Table S3). We estimated the adenylate charge and ATP to ADP ratio of the strains (Fig. 2C and D). While neither of these parameters significantly differed, a trend of increase in ATP to ADP ratio was observed in the evolved strains. This respiratory modulation in the evolved strains, enabled by TCA tailoring, can either be due to selective regulation of NADH-producing reaction or indirect effect of SDH activity reduction based redox homeostasis (24).

The absence of NDH-2 in mammalian ETS has made this enzyme an attractive target for antimicrobial development (25). The detailed examination of this compensatory OXPHOS-TCA interplay, and its multispecies applicability is critical for the effective therapeutic targeting of NDH-2.

## ACKNOWLEDGMENTS

The funders had no role in study design, data collection, and interpretation, or the decision to submit the work for publication.

## AUTHOR AFFILIATIONS

[1]Department of Biological Sciences, Tata Institute of Fundamental Research, Mumbai, Maharashtra, India

[2]Department of Bioengineering, University of California, San Diego, La Jolla, California, USA

[3]Department of Chemical Sciences, Tata Institute of Fundamental Research, Mumbai, Maharashtra, India

[4]Novo Nordisk Foundation Center for Biosustainability, Technical University of Denmark, Kemitorvet, Kongens, Lyngby, Denmark

## AUTHOR ORCIDs

Amitesh Anand  http://orcid.org/0000-0002-3720-4301

## FUNDING

| Funder | Grant(s) | Author(s) |
| --- | --- | --- |
| Tata Institute of Fundamental Research (TIFR) | 19P0120 | Amitesh Anand |
| Department of Biotechnology, Ministry of Science and Technology, India (DBT) | Ramalingaswami Fellowship | Amitesh Anand |
| Novo Nordisk Fonden (NNF) | NNF10CC1016517 and NNF20CC0035580 | Adam M. Feist |

## AUTHOR CONTRIBUTIONS

Nikita Goel, Data curation, Formal analysis, Investigation, Methodology, Visualization, Writing – original draft | Stuti Srivastav, Investigation | Arjun Patel, Formal analysis, Visualization | Akshay Shirsath, Formal analysis, Visualization | Tushar Ranjan Panda, Methodology, Resources | Malay Patra, Methodology, Resources | Adam M. Feist, Funding acquisition, Investigation | Amitesh Anand, Conceptualization, Data curation, Formal analysis, Funding acquisition, Investigation, Methodology, Project administration, Supervision, Visualization, Writing – original draft

## DATA AVAILABILITY

DNAseq data supporting this study can be accessed from the NCBI Sequence Read Archive (PRJNA1015560). RNAseq data supporting this study can be accessed from the NCBI Gene Expression Omnibus (GSE242875). The expression data for *E. coli* wild type and glucose minimal media evolved strain (GMOS) were accessed from GSE135516.

## ADDITIONAL FILES

The following material is available online.

### Supplemental Material

**Supplemental figure, tables, and methods (Spectrum02225-23-s0001.pdf).** Supplemental material.

## Open Peer Review

**PEER REVIEW HISTORY (review-history.pdf).** An accounting of the reviewer comments and feedback.

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
