## [Reviewer comments · Microbiology Spectrum]

Microbiology Spectrum

TCA cycle tailoring facilitates optimal growth of proton pumping NADH dehydrogenase-dependent *Escherichia coli*

Nikita Goel, Stuti Srivastav, Arjun Patel, Akshay Shirsath, Tushar Panda, Malay Patra, Adam Feist, and Amitesh Anand

Corresponding Author(s): Amitesh Anand, Tata Institute of Fundamental Research

Review Timeline:

Submission Date:	May 26, 2023
Editorial Decision:	June 21, 2023
Revision Received:	August 20, 2023
Editorial Decision:	September 4, 2023
Revision Received:	September 12, 2023
Accepted:	September 12, 2023

Editor: Amit Singh

Reviewer(s): The reviewers have opted to remain anonymous.

Transaction Report:

DOI: <https://doi.org/10.1128/spectrum.02225-23>

June 21, 2023

Dr. Amitesh Anand
Tata Institute of Fundamental Research
Department of Biological Sciences
Homi Bhabha Road
Colaba
Mumbai, Maharashtra 400005
India

Re: Spectrum02225-23 (**TCA cycle tailoring facilitates optimal growth of proton pumping NADH dehydrogenase-dependent *Escherichia coli***)

Dear Dr. Amitesh Anand:

Your manuscript is reviewed by three experts in the area of metabolism, redox, and evolution. Our experts recognized the potential of your manuscript, however, they have raised several concerns and suggested modifications. Please address these queries as constructively as possible.

Link Not Available

Sincerely,

Amit Singh

Journals Department
Reviewer comments:

Reviewer #1 (Comments for the Author):

The study by Goel and colleagues shed lights on the bioenergetics flexibility of the model organism *E. coli* through adaptive evolution experiment.

The study is well designed and logically organised.
The manuscript is well written and cites the appropriate literature.

However, in order to definitively validate their observations, the authors should consider:

- 1) Measure the AXP pool and determine the adenylate energy charge to understand the growth advantage observed that $e\Delta ndh-A$ in M9 as Figure 1 panel B as well as in the GMOS strain.
- 2) Perform targeted ^{13}C stable isotope metabolomics to validate the figure 1 panel G and confirm that indeed carbon flux is redirected in $e\Delta ndh-A$ compared to Δndh

Reviewer #2 (Comments for the Author):

Goel et al present an interesting observation on how *Escherichia coli* can rewire oxidative phosphorylation to adjust to the loss of type-2 dehydrogenases. The manuscript is clear, and the conclusions are generally supported by the data presented. Nevertheless, a few issues should be addressed.

1. Given that there is no complemented strain for *ndh*, and that the genome was sequenced, it would be important to show that the unevolved *ndh* does not harbor secondary mutations that may account for the observed growth defect.
2. The deletion of *sdhA* in the *ndh* background was not compared with the *e ndh* lineages with point mutations in *sdhA* (Figure 1E), but it seems like *ndh sdhA* growth does not reach WT/ *e ndh*. A comparison between *ndh sdhA* and *e ndh*, and some discussion on why the strains don't behave similarly would be important.
3. Is the difference in growth between *ndh* and *ndh sdh* significant?
4. Does *ndh sdhA* have the same adaptive regarding the *nuo* operon? This would strengthen the proposed causality.
5. One hypothesis that is not clearly stated is that the rescue effect observed in *e ndh* could be, at least in part, due to *nuo* overexpression.
6. The authors propose a "a general increase in the central carbon metabolism in the evolved strains" based on a metabolic model. In the next paragraph, the authors state "The selective silencing of the NADH-producing reactions could be an adaptive response to maintain a growth conducive redox environment." Given that the central carbon metabolism is the main generator of NADH, these statements are somewhat contradictory. It would be important to clarify which NADH producing reactions are predicted to be inhibited and which specific pathways from CCM have increased flux.

Reviewer #3 (Comments for the Author):

The manuscript "TCA cycle tailoring facilitates optimal growth of proton pumping NADH dehydrogenase-dependent *Escherichia coli*" by Goel et. al., has used adaptive laboratory evolution to study the consequence of deleting type II NADH dehydrogenase (*ndh*) on the growth and metabolism of *E. coli*. They show that mutants lacking *ndh* are defective for growth on glucose. Interestingly, the *ndh* deletion strain overcomes growth defect in 300 generations by: i) accumulating mutations that decrease the activity of succinate dehydrogenase (SDH), and ii) upregulating type I NADH dehydrogenase (*Nuo*). The authors suggest that reduction in the activity of succinate dehydrogenase optimises the growth of a Δndh strain.

Major comments:

1. Some discussion regarding how lowering of SDH activity might optimise the growth of Δndh strain is required.
2. More replicates are required for SDH activity assay. In Figure 1D, for Δndh and $e\Delta ndh-D$, 2 of the 3 replicates have SDH activity lower than the WT.
3. In Figure 1F, authors have checked the expression of components of NADH dehydrogenase I (*nuo*). The *nuo* genes are upregulated after adaptive evolution of a Δndh strain. To support the claim that in the evolved Δndh strain, SDH and *Nuo* activities are antagonistic, it will be important to compare *Nuo* activity in the evolved and unevolved Δndh strains.
4. In Supplementary figure 1A and B, data is presented with standard error of the mean. However, for all other growth curve data, data is presented with standard deviation. Both for consistency and better representation of the data, the data should be presented with standard deviation.
5. In Supplementary figure 1B, Δndh shows growth defect in succinate. Some explanation of this phenotype is required in the text: Is this because of less energy generation, redox imbalance, or due to a decrease in SDH activity (Figure 1D: 2 of the 3 replicates have SDH activity lower than the WT).
6. Page 4, Lines 129-131, the authors state that 'The selective silencing of the NADH-producing reactions could be an adaptive response to maintain a growth conducive redox environment.' Here they should comment on the evolved $\Delta ndh-D$ strain where *sucA* is mutated. *SucA* is a component of 2-oxoglutarate dehydrogenase, and is involved in the NADH generating step in TCA cycle. They should also comment on why other steps of NADH generation in TCA cycle were not affected in the evolved lineages, e.g., malate dehydrogenase.

Minor comments

1. For both main and supplementary figures depicting growth curve data, please provide details of the media used for monitoring growth of strains in the figure legends, especially since in different experiments either glucose or succinate is used as the carbon source.
2. Supplementary Figure 1A and B, Since the *AsdhA* strain fails to grow, there is no data corresponding to this strain in these figures. Hence, the label for *AsdhA* shown with the figures is misleading, and can be omitted. A mention of the phenotype in the text would be sufficient.
3. Page 4, Line 99, Provide full form for $e\Delta ndh$

Staff Comments:

Preparing Revision Guidelines

Please return the manuscript within 60 days; if you cannot complete the modification within this time period, please contact me. If you do not wish to modify the manuscript and prefer to submit it to another journal, please notify me of your decision immediately so that the manuscript may be formally withdrawn from consideration by Microbiology Spectrum.

Goel et al present an interesting observation on how *Escherichia coli* can rewire oxidative phosphorylation to adjust to the loss of type-2 dehydrogenases. The manuscript is clear, and the conclusions are generally supported by the data presented. Nevertheless, a few issues should be addressed.

1. Given that there is no complemented strain for Δndh , and that the genome was sequenced, it would be important to show that the unevolved Δndh does not harbor secondary mutations that may account for the observed growth defect.
2. The deletion of *sdhA* in the Δndh background was not compared with the $e\Delta ndh$ lineages with point mutations in *sdhA* (Figure 1E), but it seems like $\Delta ndh\Delta sdhA$ growth does not reach WT/ $e\Delta ndh$. A comparison between $\Delta ndh\Delta sdhA$ and $e\Delta ndh$, and some discussion on why the strains don't behave similarly would be important.
3. Is the difference in growth between Δndh and $\Delta ndh\Delta sdh$ significant?
4. Does $\Delta ndh\Delta sdhA$ have the same adaptive regarding the *nuo* operon? This would strengthen the proposed causality.
5. One hypothesis that is not clearly stated is that the rescue effect observed in $e\Delta ndh$ could be, at least in part, due to *nuo* overexpression.
6. The authors propose a "a general increase in the central carbon metabolism in the evolved strains" based on a metabolic model. In the next paragraph, the authors state "The selective silencing of the NADH-producing reactions could be an adaptive response to maintain a growth conducive redox environment.". Given that the central carbon metabolism is the main generator of NADH, these statements are somewhat contradictory. It would be important to clarify which NADH producing reactions are predicted to be inhibited and which specific pathways from CCM have increased flux.

Comments for the authors

The manuscript “TCA cycle tailoring facilitates optimal growth of proton pumping NADH dehydrogenase-dependent *Escherichia coli*” by Goel *et. al.*, has used adaptive laboratory evolution to study the consequence of deleting type II NADH dehydrogenase (*ndh*) on the growth and metabolism of *E. coli*. They show that mutants lacking *ndh* are defective for growth on glucose. Interestingly, the *ndh* deletion strain overcomes growth defect in 300 generations by: i) accumulating mutations that decrease the activity of succinate dehydrogenase (SDH), and ii) upregulating type I NADH dehydrogenase (Nuo). The authors suggest that reduction in the activity of succinate dehydrogenase optimises the growth of a Δndh strain.

Major comments:

1. Some discussion regarding how lowering of SDH activity might optimise the growth of Δndh strain is required.
2. More replicates are required for SDH activity assay. In Figure 1D, for Δndh and $e\Delta ndh-D$, 2 of the 3 replicates have SDH activity lower than the WT.
3. In Figure 1F, authors have checked the expression of components of NADH dehydrogenase I (*nuo*). The *nuo* genes are upregulated after adaptive evolution of a Δndh strain. To support the claim that in the evolved Δndh strain, SDH and Nuo activities are antagonistic, it will be important to compare Nuo activity in the evolved and unevolved Δndh strains.
4. In Supplementary figure 1A and B, data is presented with standard error of the mean. However, for all other growth curve data, data is presented with standard deviation. Both for consistency and better representation of the data, the data should be presented with standard deviation.
5. In Supplementary figure 1B, Δndh shows growth defect in succinate. Some explanation of this phenotype is required in the text: Is this because of less energy generation, redox imbalance, or due to a decrease in SDH activity (Figure 1D: 2 of the 3 replicates have SDH activity lower than the WT).
6. Page 4, Lines 129-131, the authors state that ‘The selective silencing of the NADH-producing reactions could be an adaptive response to maintain a growth conducive redox environment.’ Here they should comment on the evolved $\Delta ndh-D$ strain where *sucA* is mutated. *SucA* is a component of 2-oxoglutarate dehydrogenase, and is involved in the NADH generating step in TCA cycle. They should also comment on why other steps of NADH generation in TCA cycle were not affected in the evolved lineages, e.g., malate dehydrogenase.

Minor comments

1. For both main and supplementary figures depicting growth curve data, please provide details of the media used for monitoring growth of strains in the figure legends, especially since in different experiments either glucose or succinate is used as the carbon source.

2. Supplementary Figure 1A and B, Since the $\Delta sdhA$ strain fails to grow, there is no data corresponding to this strain in these figures. Hence, the label for $\Delta sdhA$ shown with the figures is misleading, and can be omitted. A mention of the phenotype in the text would be sufficient.
3. Page 4, Line 99, Provide full form for $e\Delta ndh$

August 20, 2023

Manuscript No.: Spectrum02225-23

Dear Editor,

We are very excited to submit the revised version of our manuscript, "*TCA cycle tailoring facilitates optimal growth of proton pumping NADH dehydrogenase-dependent Escherichia coli*," which, we believe, has benefited significantly from the review process. We would like to thank the reviewers for their time and effort in providing a constructive evaluation and insightful review of the manuscript. Our response to each remark is presented below.

Reviewers' Comments: **Black Text**

Author's Response: **Blue Text**

Reviewer #1:

The study by Goel and colleagues shed light on the bioenergetics flexibility of the model organism *E. coli* through an adaptive evolution experiment.

The study is well-designed and logically organized. The manuscript is well-written and cites the appropriate literature.

Thank you for your appreciation and the encouraging review!

However, in order to definitively validate their observations, the authors should consider:

1) Measure the AXP pool and determine the adenylate energy charge to understand the growth advantage observed that $e\Delta ndh-A$ in M9 as Figure 1 panel B as well as in the GMOS strain.

We performed the adenylate charge calculations, but the values were not significantly different between evolved vs unevolved strains. We further went on to calculate ATP to ADP ratios. Although ATP:ADP ratios were also not significantly different, we observed a trend towards an increase in the ratio upon evolution.

The adenylyate charge and ATP/ADP ratios were calculated for WT, GMOS, Δndh , e Δndh -A, and $\Delta ndh\Delta sdhA$ using UPLC. A total of five biological replicates were used for each strain. The solid bar for each strain in the violin plot represents the median while the dotted lines represent the lower and upper quartiles.

We noticed that the adenylyate charge is reported to be unaffected by growth rate and several other perturbations (1–3). Here, the growth advantage upon evolution appears to coincide with a higher ATP:ADP ratio.

2) Perform targeted ^{13}C stable isotope metabolomics to validate the figure 1 panel G and confirm that indeed carbon flux is redirected in e Δndh -A compared to Δndh

Our flux map for the central carbon metabolism uses a ME-model or a model of metabolism and expression. These models are capable of predicting the optimal macromolecular (protein, nucleotide, cofactor) expression required for growth. Our models are constrained based on the exchange rates of several key metabolites like glucose, acetate, lactate, succinate, formate, and ethanol. Notably, succinate, formate, and ethanol were below the detection limit. We are also constraining the model with measured growth rates and expression data. Thus, we can be more confident in the simulated metabolic fluxes as opposed to just using the HPLC data or more basic metabolic models.

Having said that, we recognize the value of the suggested approach, but ^{13}C -based metabolomics is outside our experimental expertise, and we would like to be excused from this line of experimentation for this case.

Reviewer #2:

Goel et al. present an interesting observation on how *Escherichia coli* can rewire oxidative phosphorylation to adjust to the loss of type-2 dehydrogenases. The manuscript is clear, and the conclusions are generally supported by the data presented.

We always attempt to produce lucid manuscripts and are glad that our efforts are recognizable. Nevertheless, a few issues should be addressed.

1. Given that there is no complemented strain for *ndh*, and that the genome was sequenced, it would be important to show that the unevolved *ndh* does not harbor secondary mutations that may account for the observed growth defect.

Before the start of our laboratory evolution experiments, we carefully examine the starting strain. As noted by you, we sequenced the whole genome of the strain. We have created the largest database of mutations from adaptive laboratory evolution experiments, and we refer to this database (ALEdb) to ensure the validity of strains (4). We did not find any mutations in the unevolved Δndh that are consequential in nature.

2. The deletion of *sdhA* in the *ndh* background was not compared with the *endh* lineages with point mutations in *sdhA* (Figure 1E), but it seems like *ndhsdhA* growth does not reach WT/*endh*. A comparison between *ndhsdhA* and *endh*, and some discussion on why the strains don't behave similarly would be important.

Yes, $\Delta ndh\Delta sdhA$ does not reach *e* Δndh growth but their growth overlaps with WT.

Growth curve of WT, Δndh , *e* Δndh -A, and $\Delta ndh\Delta sdhA$ on M9 minimal media with glucose. The data represent a mean of five biological replicates (with three technical replicates each) and the error bars show the standard error of mean.

The evolved Δndh lineages have additional mutations responsible for the growth optimization on minimal media in an aerobic environment. Two replicates have mutations that

relieve a defect in pyrimidine biosynthesis caused by a 1-bp deletion in the *rph* gene. Mutations in this genomic region are common in *E. coli* K12 MG1655 when the strain adapts to minimal media (4, 5). The other two lineages have mutation in a putative RNase adaptor protein, *yicC*. We verified the impact of this mutation and observed that *yicC* deletion improves the growth of $\Delta ndh\Delta sdhA$ strain. YicC regulates a small regulatory RNA, RhyB, that modulates the central carbon metabolism and redox homeostasis (6, 7). Notably, *yicC* is immediately upstream of the *rph-pyrE* region and the 103 nucleotide deletion observed in this gene may again be improving the growth through optimization of *pyrE* expression.

Growth curve of Δndh , $\Delta ndh\Delta yicC$, $\Delta ndh\Delta sdhA$ and $\Delta ndh\Delta sdhA\Delta yicC$ on M9 minimal media with glucose. The data represent a mean of five biological replicates (with three technical replicates each) and the error bars show the standard error of mean.

Also, there is a chance that the quantitative impact of the point mutations observed in the evolved strains differ from the complete deletion of the gene.

3. Is the difference in growth between *ndh* and *ndhsdh* significant?

We performed statistical analysis and observed the growth rates to differ significantly between Δndh and $\Delta ndh\Delta sdhA$.

Growth rates of Δndh and $\Delta ndh\Delta sdhA$. For each strain, 5 biological replicates are used for growth rate calculations. The solid line inside the box plots represents the median while the whiskers represent the maximum and minimum data points. * P value < 0.05, two-tailed Mann-Whitney test.

4. Does *ndhsdhA* have the same adaptive regarding the *nuo* operon? This would strengthen the proposed causality.

Our current understanding is based on growth profiling, but we do not have RNAseq data for $\Delta ndh\Delta sdhA$ strain. Your suggestion motivated us to look for some alternate explanations as well. Interestingly, the inhibition of flux through SDH (complex II) is reported to prevent ROS production at NDH-1 (complex I) by minimizing the production of metabolites contributing to reverse electron flux (8).

We intend to follow up our *Observation* article with a detailed work looking into this aspect and a characterization of every TCA reaction in the background of *ndh* deletion.

5. One hypothesis that is not clearly stated is that the rescue effect observed in *endh* could be, at least in part, due to *nuo* overexpression.

Thank you for this observation. We mentioned this observation in a subtle manner. We have now elaborated on this possibility further.

6. The authors propose a "a general increase in the central carbon metabolism in the evolved strains" based on a metabolic model. In the next paragraph, the authors state "The selective silencing of the NADH-producing reactions could be an adaptive response to maintain a growth conducive redox environment.". Given that the central carbon metabolism is the main generator of NADH, these statements are somewhat contradictory. It would be important to clarify which NADH producing reactions are predicted to be inhibited and which specific pathways from CCM have increased flux.

The hypothesis about selective silencing of NADH-producing reactions is based on an Escher-based examination of the impact of SDH (SUCD) and AKGDH deletion on the TCA and

the ME-model based flux simulation (9). Both deletions appear to reduce flux through the AKGDH-catalyzed reaction. We are motivated to probe this hypothesis deeper.

Flux through central carbon metabolism network simulated using Escher-FBA in the WT and *in-silico* KOs of succinate dehydrogenase (SDH/SUCD) and alpha-ketoglutarate dehydrogenase (AKGDH). Highlighted reactions are as follows (1) Glyceraldehyde 3- phosphate dehydrogenase, (2) Pyruvate Dehydrogenase, (3) AKGDH, (4) Isocitrate dehydrogenase, and (5) Malate dehydrogenase. The colorscale represent flux values with intensity of red color showing increase.

We do observe a similar flux depletion through AKGDH in our data based flux simulation.

Reviewer #3 (Comments for the Author):

The manuscript "TCA cycle tailoring facilitates optimal growth of proton pumping NADH dehydrogenase-dependent *Escherichia coli*" by Goel et. al., has used adaptive laboratory evolution to study the consequence of deleting type II NADH dehydrogenase (*ndh*) on the growth and metabolism of *E. coli*. They show that mutants lacking *ndh* are defective for growth on glucose. Interestingly, the *ndh* deletion strain overcomes growth defect in 300 generations by: i) accumulating mutations that decrease the activity of succinate dehydrogenase (SDH), and ii) upregulating type I NADH dehydrogenase (Nuo). The authors suggest that reduction in the activity of succinate dehydrogenase optimises the growth of a Δndh strain.

Thank you for the overview of our work.

Major comments:

1. Some discussion regarding how lowering of SDH activity might optimize the growth of Δndh strain is required.

Among the two NADH-dehydrogenases of *E. coli*, NDH-2 (*ndh*) has a higher catalytic turnover for NADH oxidation. In Δndh , the oxidation of NADH might get compromised. Since the TCA cycle contributes significantly to NADH production, lowering flux through NADH producing reactions can help mitigate this redox imbalance. This can explain the presence of *sdhA* mutations in the evolved strains. Additionally, the inhibition of flux through SDH is reported to prevent ROS production at NDH-1 by minimizing the production of metabolites contributing to reverse electron flux (8). We have elaborated these possibilities in the revised manuscript.

2. More replicates are required for SDH activity assay. In Figure 1D, for Δndh and e Δndh -D, 2 of the 3 replicates have SDH activity lower than the WT.

We have generated more replicates and hope you will find the revised data satisfactory.

In vitro SDH activity of $\Delta sdhA$, WT, Δndh (unevolved and evolved) strains. All seven replicate values are displayed with standard error. A significance test using one-way ANOVA was performed compared to $\Delta sdhA$.

3. In Figure 1F, authors have checked the expression of components of NADH dehydrogenase I (nuo). The nuo genes are upregulated after adaptive evolution of a Δ ndh strain. To support the claim that in the evolved Δ ndh strain, SDH and Nuo activities are antagonistic, it will be important to compare Nuo activity in the evolved and unevolved Δ ndh strains.

Our proposition is based on the increase in the expression of *nuo* genes, as there is no change in the protein sequence. We attempted to perform the Nuo activity assay but could not get presentable data (10) (11)

4. In Supplementary figure 1A and B, data is presented with standard error of the mean. However, for all other growth curve data, data is presented with standard deviation. Both for consistency and better representation of the data, the data should be presented with standard deviation.

This difference was due to the use of two different platforms for growth profiling (micro-well plate based assay and culture-tube based assay). In micro-well plate based assay we had three technical replicates for each biological replicate, thus we presented standard error of mean. Whereas culture-tube based assay had biological replicates only and, therefore, we presented standard deviation. We have re-generated all the data using micro-well plate based assay read using plate reader to ensure uniformity across the manuscript.

5. In Supplementary figure 1B, Δ ndh shows growth defect in succinate. Some explanation of this phenotype is required in the text: Is this because of less energy generation, redox imbalance, or due to a decrease in SDH activity (Figure 1D: 2 of the 3 replicates have SDH activity lower than the WT).

Δ ndh shows growth defects on both glucose and succinate. As compared to glucose, succinate is more of a respiratory metabolite than fermentative. This respiro-fermentative bias will exert extra pressure on ETS functioning in Δ ndh growing on succinate. The growth defect on succinate could be for the same reason as that for glucose. Also, Δ ndh has reduced SDH activity compared to WT, which is potentially aiding the growth defect of Δ ndh on succinate. We have elaborated this part in the revised manuscript.

6. Page 4, Lines 129-131, the authors state that 'The selective silencing of the NADH-producing reactions could be an adaptive response to maintain a growth conducive redox environment.' Here they should comment on the evolved Δ ndh-D strain where *sucA* is mutated. *SucA* is a component of 2-oxoglutarate dehydrogenase, and is involved in the NADH generating step in TCA cycle. They should also comment on why other steps of NADH generation in TCA cycle were not affected in the evolved lineages, e.g., malate dehydrogenase.

A similar clarification has been sought by the second reviewer. We are copying the response here with additional thoughts for easy reference-

The hypothesis about selective silencing of NADH-producing reactions is based on Escher based examination of the impact of SDH (SUCD) and AKGDH deletion on the TCA and the ME-model based flux simulation. Both deletions appear to reduce flux through the AKGDH catalyzed reaction.

Flux through the central carbon metabolism network simulated using Escher-FBA in the WT and *in-silico* KOs of succinate dehydrogenase (SDH/SUCD) and alpha-ketoglutarate dehydrogenase (AKGDH). Highlighted reactions are as follows (1) Glyceraldehyde 3- phosphate dehydrogenase, (2) Pyruvate Dehydrogenase, (3) AKGDH, (4) Isocitrate dehydrogenase, and (5) Malate dehydrogenase.

We do not have a clear explanation for why other NADH producing reactions were not targeted for adaptation. The answer might lie in the metabolic inter-connections or simply adding more lineages in the future evolution experiment. We are motivated to probe this hypothesis deeper.

Minor comments

1. For both main and supplementary figures depicting growth curve data, please provide details of the media used for monitoring growth of strains in the figure legends, especially since in different experiments either glucose or succinate is used as the carbon source.

Appropriate details are added in the revised manuscript.

2. Supplementary Figure 1A and B, Since the Δ sdhA strain fails to grow, there is no data corresponding to this strain in these figures. Hence, the label for Δ sdhA shown with the figures is misleading, and can be omitted. A mention of the phenotype in the text would be sufficient.

There is data for Δ sdhA in the plot. The strain did not show an increase in OD and essentially is getting masked with $e\Delta$ ndh-A. We have now plotted the time-separated alternate OD values for Δ sdhA and $e\Delta$ ndh-A so that both data can be noticed in the plot.

3. Page 4, Line 99, Provide full form for eAndh

We have added a scheme in the revised manuscript to elaborate on the labeling scheme.

We once again thank the reviewers for their valuable insights and suggestions. This has significantly strengthened the manuscript.

References:

1. Vasilakou E, van Loosdrecht MCM, Wahl SA. 2020. Escherichia coli metabolism under short-term repetitive substrate dynamics: adaptation and trade-offs. *Microb Cell Fact* 19:1–19.
2. Adenylate Energy Charge in Escherichia coli During Growth and Starvation. <http://dx.doi.org/10.1128/jb.108.3.1072-1086.1971>. Retrieved 18 August 2023.
3. 2022. Homeostasis of the biosynthetic E. coli metabolome. *iScience* 25:104503.
4. Phaneuf PV, Gosting D, Palsson BO, Feist AM. 2018. ALEdb 1.0: a database of mutations from adaptive laboratory evolution experimentation. *Nucleic Acids Res* 47:D1164–D1171.
5. LaCroix RA, Sandberg TE, O'Brien EJ, Utrilla J, Ebrahim A, Guzman GI, Szubin R, Palsson BO, Feist AM. 2014. Use of Adaptive Laboratory Evolution To Discover Key Mutations Enabling Rapid Growth of Escherichia coli K-12 MG1655 on Glucose Minimal Medium. *Appl Environ Microbiol* <https://doi.org/10.1128/AEM.02246-14>.
6. Lyu Y, Wu J, Shi Y. 2019. Metabolic and physiological perturbations of Escherichia

coli W3100 by bacterial small RNA RyhB. *Biochimie* 162:144–155.

7. Chen J, To L, de Mets F, Luo X, Majdalani N, Tai C-H, Gottesman S. 2021. A fluorescence-based genetic screen reveals diverse mechanisms silencing small RNA signaling in. *Proc Natl Acad Sci U S A* 118.
8. Mathers KE, Staples JF. 2019. Differential posttranslational modification of mitochondrial enzymes corresponds with metabolic suppression during hibernation. *Am J Physiol Regul Integr Comp Physiol* 317:R262–R269.
9. King ZA, Dräger A, Ebrahim A, Sonnenschein N, Lewis NE, Palsson BO. 2015. Escher: A Web Application for Building, Sharing, and Embedding Data-Rich Visualizations of Biological Pathways. *PLoS Comput Biol* 11:e1004321.
10. Hreha TN, Foreman S, Duran-Pinedo A, Morris AR, Diaz-Rodriguez P, Andrew Jones J, Ferrara K, Bourges A, Rodriguez L, Koffas MAG, Hahn M, Hauser AR, Barquera B. 2021. The three NADH dehydrogenases of *Pseudomonas aeruginosa*: Their roles in energy metabolism and links to virulence. *PLoS One* 16:e0244142.
11. 1989. Purification of NADH-ferricyanide dehydrogenase and NADH-quinone reductase from *Escherichia coli* membranes and their roles in the respiratory chain. *Biochimica et Biophysica Acta (BBA) - Bioenergetics* 977:62–69.

September 4, 2023

Dr. Amitesh Anand
Tata Institute of Fundamental Research
Department of Biological Sciences
Homi Bhabha Road
Colaba
Mumbai, Maharashtra 400005
India

Re: Spectrum02225-23R1 (**TCA cycle tailoring facilitates optimal growth of proton pumping NADH dehydrogenase-dependent *Escherichia coli***)

Dear Dr. Amitesh Anand:

In the figure legends pertaining to growth curve experiments, "minimal media" should be replaced with "minimal medium" since one carbon source condition has been used in an individual experiment.

Thank you for submitting your manuscript to Microbiology Spectrum. As you will see your paper is very close to acceptance. Please modify the manuscript along the lines I have recommended. As these revisions are quite minor, I expect that you should be able to turn in the revised paper in less than 30 days, if not sooner. If your manuscript was reviewed, you will find the reviewers' comments below.

When submitting the revised version of your paper, please provide (1) point-by-point responses to the issues raised by the reviewers as file type "Response to Reviewers," not in your cover letter, and (2) a PDF file that indicates the changes from the original submission (by highlighting or underlining the changes) as file type "Marked Up Manuscript - For Review Only". Please use this link to submit your revised manuscript. Detailed instructions on submitting your revised paper are below.

Link Not Available

Sincerely,

Amit Singh

Reviewer comments:

Reviewer #1 (Comments for the Author):

The authors have addressed the concerns.

Reviewer #2 (Comments for the Author):

The authors adequately responded to all my concerns.

Reviewer #3 (Comments for the Author):

The authors have carefully addressed the comments and incorporated these in the revised manuscript.

Minor comment:

In the figure legends pertaining to growth curve experiments, "minimal media" should be replaced with "minimal medium" since one carbon source condition has been used in an individual experiment.

Preparing Revision Guidelines

Please return the manuscript within 60 days; if you cannot complete the modification within this time period, please contact me. If you do not wish to modify the manuscript and prefer to submit it to another journal, please notify me of your decision immediately so that the manuscript may be formally withdrawn from consideration by Microbiology Spectrum.

September 11, 2023

Manuscript No.: Spectrum02225-23R1

Dear Editor,

We are glad to note that all three reviewers found our revised manuscript satisfactory. We are pleased to submit the revised manuscript (Spectrum02225-23R1) titled "TCA cycle tailoring facilitates optimal growth of proton pumping NADH dehydrogenase-dependent *Escherichia coli*," with the suggested typographical change replacing media with medium.

Reviewers' Comments: **Black Text**

Author's Response: **Blue Text**

Reviewer #1:

The authors have addressed the concerns.

We appreciate your help in improving the manuscript.

Reviewer #2:

The authors adequately responded to all my concerns.

Your comments helped us improve our manuscript.

Reviewer #3 (Comments for the Author):

The authors have carefully addressed the comments and incorporated these in the revised manuscript.

Minor comment:

In the figure legends pertaining to growth curve experiments, "minimal media" should be replaced with "minimal medium" since one carbon source condition has been used in an individual experiment.

Thank you for your detailed review. We are glad to have addressed your comments.

September 12, 2023

Dr. Amitesh Anand
Tata Institute of Fundamental Research
Department of Biological Sciences
Homi Bhabha Road
Colaba
Mumbai, Maharashtra 400005
India

Re: Spectrum02225-23R2 (**TCA cycle tailoring facilitates optimal growth of proton pumping NADH dehydrogenase-dependent *Escherichia coli***)

Dear Dr. Amitesh Anand:

Your manuscript has been accepted, and I am forwarding it to the ASM Journals Department for publication. You will be notified when your proofs are ready to be viewed.

Sincerely,

Amit Singh
Editor, Microbiology Spectrum
